# New Perspectives on Sleep Regulation by Tea: Harmonizing Pathological Sleep and Energy Balance under Stress

**DOI:** 10.3390/foods11233930

**Published:** 2022-12-05

**Authors:** Jin Ouyang, Yuxuan Peng, Yushun Gong

**Affiliations:** 1Key Laboratory of Tea Science of Ministry of Education, Changsha 410128, China; 2National Research Center of Engineering and Technology for Utilization of Botanical Functional Ingredients, Changsha 410128, China; 3Key Laboratory for Evaluation and Utilization of Gene Resources of Horticultural Crops, Ministry of Agriculture and Rural Affairs of China, Hunan Agricultural University, Changsha 410128, China; 4College of Physical Education, Hunan City University, Yiyang 413002, China

**Keywords:** sickness sleep, tea, energy metabolism, damage repair, energy homeostasis

## Abstract

Sleep, a conservative evolutionary behavior of organisms to adapt to changes in the external environment, is divided into natural sleep, in a healthy state, and sickness sleep, which occurs in stressful environments or during illness. Sickness sleep plays an important role in maintaining energy homeostasis under an injury and promoting physical recovery. Tea, a popular phytochemical-rich beverage, has multiple health benefits, including lowering stress and regulating energy metabolism and natural sleep. However, the role of tea in regulating sickness sleep has received little attention. The mechanism underlying tea regulation of sickness sleep and its association with the maintenance of energy homeostasis in injured organisms remains to be elucidated. This review examines the current research on the effect of tea on sleep regulation, focusing on the function of tea in modulating energy homeostasis through sickness sleep, energy metabolism, and damage repair in model organisms. The potential mechanisms underlying tea in regulating sickness sleep are further suggested. Based on the biohomology of sleep regulation, this review provides novel insights into the role of tea in sleep regulation and a new perspective on the potential role of tea in restoring homeostasis from diseases.

## 1. Introduction

Sleep, a spontaneous and reversible resting state in mammals, is essential in optimizing energy conservation or allocation, regulating core molecular and cellular processes, and enhancing brain functions [1,2,3]. Both quality and quantity are fundamental components of sleep. Poor sleep quality contributes to chronic diseases such as kidney disease, hypertension, obesity, and diabetes mellitus [4,5,6]. Meanwhile, lack of sleep can induce substantial short- and long-term memory impairment and is a risk factor for anxiety and depression [7,8,9,10].

There are two types of sleep: natural sleep and sickness sleep. Sickness sleep is the sensation of being sleepy and fatigued under stress, infection, or disease conditions. In lower animals, sickness sleep, known as stress-induced sleep (SIS), has a non-negligible role in supporting recovery from injury [11]. In *Caenorhabditis elegans*, ALA neuron-dependent SIS is important to increase survival after cellular stress [12]. Increasing sleep increases survival after oxidative challenge in *Drosophila* [13]. Sleep plays an equal role in recovery from exhaustion and illness in higher organisms. In mice, deeper resting leads to a faster recovery [14]. In a symptom assessment of 59 burnt-out employees taking extended sick leave, sleep was found to play an important role both in symptom improvement and in return to work [15].

Energy homeostasis is essential for the normal physiological activities of the body. Damage repair is induced by stress and promotes sleep; however, it also results in a disequilibrium of energy homeostasis due to increased energy requirements. Cellular energy charge and glycogen levels, induced by energy metabolism, may operate as signaling mechanisms. This may be when energy stores reallocate energy away from high-energy demands and gene expression during the waking state toward biological processes favoring sleep [16].

As one of the most widely consumed non-alcoholic beverages, tea has numerous recognized health benefits, such as resistance to stress and sleep regulation [17,18]. Tea effectively improves resistance to various stresses, including heat, oxidative, and ultraviolet (UV) stress [19,20,21]. Oolong tea consumption has been reported to improve stress symptoms and alleviate the elevated plasma lipid peroxidation levels caused by nighttime stress in study subjects [22]. Green tea has anti-stress effects, in which theanine, Epigallocatechin (EGC), and arginine synergistically eliminate the antagonistic effects of caffeine and Epigallocatechin gallate (EGCG) on adrenal hypertrophy induced by psychosocial stress in mice [23]. Sleep duration and quality are affected by the frequency and amount of tea consumed [24,25,26,27,28]. A recent study that investigated the effects of tea on alleviating acute alcohol intoxication (AAI) in mice found that tea reduced AAI and regulated sleep by inhibiting oxidative stress and inflammation [29].

Obesity and diabetes are caused by the imbalance of the body’s energy homeostasis. Tea has been found to reduce obesity and ease diabetes. Oolong tea extract reduces lipid accumulation by regulating lipid metabolism and intestinal flora distribution, thus inhibiting high-diet-induced obesity in mice [30]. Green tea components are involved in protein interactions and cell signaling pathways that regulate energy metabolism, including glycogen synthesis and glucose reabsorption, thereby reducing glycogen accumulation and improving the pathological features in diabetic mice [31].

Previous studies have illustrated the role of tea in regulating stress injury response, healthy sleep, and energy homeostasis; whether and how tea regulates sickness sleep under stress or illness remains unclear. Therefore, this review covers recent discoveries on the effect of tea on sickness sleep and explores whether the chemical properties and consumption of tea can influence energy homeostasis and sickness sleep in stressful environments. This review also explores the potential link between tea consumption and the effects of sleep under stress.

## 2. Tea and Sleep

### 2.1. Tea

Tea has originated in China and since spread worldwide. People in more than 160 countries and regions habitually drink tea. Based on the degree of fermentation during processing, tea is classified into six types: green, yellow, white, oolong, dark, and black. Tea is rich in active ingredients, including polyphenols, amino acids, alkaloids, aromatic substances, and sugars [32]. Table 1 lists the degree of fermentation, key processes, and recent studies related to the respective health benefits of tea. It is worth noting that despite the different processing processes of the six tea types, the health benefits are all centered on their antibacterial and antiviral activities; neuroprotection; protection against cancer, obesity, diabetes, and cardiovascular diseases; and sleep regulation [33].

### 2.2. Sleep

Sleep is primarily classified as either natural or sickness, of which natural sleep is regulated by the interaction of circadian and homeostatic mechanisms [11]. In mammals and avian species, natural sleep is identified based on altered brain electrical activity, recorded on an electroencephalogram (EEG); it comprises non-rapid eye movement sleep (NREMS) and rapid eye movement sleep (REMS) [54]. During NREMS, the EEG pattern shows slow oscillations, referred to as slow waves, and the muscle tone and brain activity are at rest [55,56,57]. Conversely, REMS (also termed fast-wave sleep) is characterized by reduced amplitude and faster frequency in cortical EEG [56]. Mammals breathe irregularly and with relaxed muscles in REMS, which have specific effects on the respiratory system and motor neurons [58,59].

As non-mammalians do not have differentiated EEG, sleep is usually defined according to the following behavioral states: spontaneous circadian motion quiescence, decreased reactivity, increased arousal threshold, and rapid reversibility [60,61]. In zebrafish, the sleep state is characterized by a reversible circadian rhythm and an increased arousal threshold [62,63]. In *Drosophila*, sleep is typically characterized as an inactive period spanning 5 min or more and heightened arousal thresholds [64,65]. Developmentally timed sleep (DTS) in *C. elegans* is the 2–3 h of quiescence during the transition between larval stages [60].

### 2.3. Sickness Sleep

Sickness sleep occurs in stressful environments or diseases as a sensation of being sleepy and fatigued. Sleepiness is common following traumatic injury, particularly traumatic brain injury, with more severe injuries resulting in greater sleepiness in humans [66]. Inflammation has been suggested as a potential contributor to the development of fatigue [67]. Stress activates sleep-related brain regions and induces sleep-like inactivity in mice [68]. *Murine gammaherpesvirus* 68 alters sleep, activity, and temperature in a manner suggestive of fatigue in mice [69]. Sickness sleep is easier to induce in non-mammals and is called SIS. The sleep duration in *Drosophila melanogaster* increases following aseptic injury [70]. Long-term radiofrequency radiation exposure enhances the heat stress response and affects the expression of the circadian clock and neurotransmitter genes, thereby prolonging sleep duration [71]. SIS in *C. elegans* can be triggered by conditions of cellular stress, including noxious heat, cold, hyperosmolarity, ultraviolet irradiation, and mechanical injury [72], and exhibits behavioral quiescence similar to DTS [73]. Sleep is a conserved evolutionary behavior of organisms, whether mammals or non-mammals, to adapt to changes in the external environment [74].

## 3. Tea Effects on Sleep

Previous studies have found that people suffer from insomnia and fatigue after consuming a high intake of green tea [75]. Short sleep duration was associated with a higher intake of black tea [76]. However, other studies have suggested that tea can calm nerves and promote sleep [25,77]. For example, after consuming black tea with γ-aminobutyric acid (GABA), the sleeping time with sodium pentobarbital was significantly prolonged, and the quality was improved in mice [78]. Green tea extracts can improve sleep disturbances and stabilize mood in humans [79,80]. Fragrant compounds in oolong tea have tranquilizing effects on the brains of mice [81]. Different active ingredients in tea have various implications for sleep (as shown in Figure 1).

Tea polyphenols play an essential role in regulating sleep and mood [82]. After consuming polyphenols, sleep duration is prolonged, and the quality is significantly improved in humans [83]. Meanwhile, tea polyphenols treatment effectively enhances cognitive impairment, memory impairment, and anxiety-like behaviors in sleep-disordered mice [82,84].

Theanine, the main amino acid in tea, promotes sleep. Studies have shown that following treatment with L-theanine, the Pittsburgh Sleep Quality Index (PSQI) subscale scores for sleep latency and satisfaction are improved in participants with no major psychiatric illness [85,86]. L-theanine increases sleep duration and shorten sleep latency in mice [87], while L-theanine/GABA mixture significantly increases NREMS and REMS in the mice [88].

Caffeine, the primary alkaloid in tea, is used to combat high sleep pressures. After 10 days of caffeine intake, a significant reduction in the gray matter volume in the medial temporal lobe was observed [89]. Oral caffeine administration significantly increases sleep latency and decreases the amount of NREMS in mice [90].

Tea contains various aromatic substances. Although natural aromatic components are not in high concentration in tea, they can be used as sleep aids [91]. One of the aromatic substances abundantly present in tea is linalool, which has the aroma of lily or magnolia flowers, and Linalyl acetate has the aroma of green lemons. Studies have shown that essential oils composed of linalool and linalyl acetate can significantly reduce sleep latency and prolong sleep duration in mice [92]. Jasmine lactone, an aromatic substance, constitutes oolong tea’s floral and fruity aroma. Studies have shown that inhalation of *cis*-jasmone or methyl jasmonate significantly increases the sleeping time in mice induced by pentobarbital and has a tranquilizing effect on their brains [81].

## 4. Does Tea Affect Sickness Sleep and Maintain Energy Homeostasis?

The relationship between sleep and energy is well-discussed, and the energy allocation model proposes that animal species share a universal sleep function. Sleep conserves energy by (i) reducing energy requirements for core thermoregulatory defenses and skeletal muscle tone when the external environment changes and (ii) enhancing energy appropriation for somatic and CNS-related processes [16,93,94].

Energy homeostasis in animals is maintained through energy metabolism via a dynamic balance between consumption and supply. However, animals in complex environments are subjected to daily stresses, such as solar radiation, pathogenic bacterial infections, and mechanical damage. Therefore, they suffer from organismal damage and must increase their energy expenditure for damage repair, thus breaking the energy balance for a short period. Therefore, animals have evolved sickness sleep behaviors to maintain energy homeostasis in stressful environments, a strategy for conserving energy consumption and optimizing energy allocation [2,54,74,95].

Many studies have shown that tea effectively improves stress resistance, including heat, oxidative, and UV [19,20,21]. Previous research in the laboratory found that an aqueous extract of black tea promoted SIS and prolonged the lifespan of *C. elegans* compared with the control group under ultraviolet irradiation. This suggests a link between lifespan and SIS, in which tea may play a role. However, the mechanism by which tea regulates sickness sleep remains unclear. In this review, a possible mechanism for this phenomenon is proposed. Tea may promote sickness sleep and maintain energy homeostasis, stimulating the organism’s health by acting on damage repair and energy metabolism (Figure 2).

### 4.1. Tea Acts on the Brain–Gut Axis to Regulate Sickness Sleep

The active substances of tea are known to act on the brain–gut axis to regulate sleep [96]. This review summarizes the relevant research and hypotheses regarding tea regulation of sickness sleep under stress via the brain–gut axis (Figure 3).

#### 4.1.1. Tea Acts on the Nervous System to Regulate Sickness Sleep

The nervous system is mainly composed of neurons and glia wrapped around neurons. Neurotransmitters are released between synapses to convey information that controls sleep behavior. Neurons regulate sleep and waking [95,97]. Two types of neurons that control sleep have been identified: sleep-active neurons, such as GABAergic/peptidergic neurons in the preoptic area of mammals, and sleep-promoting neurons, such as RIM and PVC neurons in *C. elegans* [56,98]. Tea effectively protects neuronal tissues and promotes growth and differentiation under stressful conditions. A previous study showed that oolong tea extracts had a protective effect against the death of neuronal cells (Neuro-2a and HT22) [99]. This is because oolong tea extracts reduce the accumulation of intracellular reactive oxygen species (ROS) and induce gene expression of cellular antioxidant enzymes. These extracts also increased average neurite length in Neuro-2a cells. Another study found that theaflavins enhanced PC12 cell survival following H_2_O_2_-induced toxicity and increased cell viability [45]. This study suggested that the neuroprotective effects of theaflavins against oxidative stress in PC12 cells are derived from the suppression of oxidant enzyme activity. Furthermore, green tea protects against hippocampal neuronal apoptosis by inhibiting the JNK/MLCK pathway [100]. The catechins EGCg and GCg effectively protect nerve cells against H_2_O_2_ or Aβ_1-42_-induced injury [101,102]. EGCg and its degradation products induce neuronal differentiation and neurite outgrowth by upregulating synaptophysin gene expression and reducing DNA methylation [103,104]. Water-soluble flavonoids enhance the neuronal differentiation of neural stem cells in a dose-dependent manner and significantly enhance neurite growth in mice [105].

Glial cells, including astrocytes, oligodendrocytes, and microglia, have neurotransmitter receptors and ion channels wrapped in neurons and play an essential role in regulating behaviors, such as movement and sleep [106]. Astroglial calcium activity changes dynamically across vigilance states and is proportional to sleep requirements. Astrocytic Gi- and Gq-coupled G-protein-coupled receptor signaling controls NREMS depth and duration, respectively [107,108,109]. Depleting microglial cells disrupts the brain tissue’s circadian rhythmicity, increases the duration of NREMS, and reduces hippocampal excitatory neurotransmission in mice [110,111]. Tea protects glial cells under stress. The pathological activation of astrocytes and other glia in mice was inhibited by green tea extract [112,113]. Chlorogenic acid increased ω-7 palmitoleic fatty acid, which was associated with an IL-6 decrease, and effectively alleviated inflammation of glial cells in mice [113,114].

Neurotransmitters mainly include acetylcholine, monoamines (dopamine and serotonin), amino acids (excitatory transmitters, such as glutamate, and inhibitory transmitters, such as GABA), and neuropeptides. Multiple neurotransmitters regulate the sleep-wake cycle [115]. FMRFamide (Phe-Met-Arg-Phe-NH2), also known as FMRFamide-related neuropeptides, and their receptors play a conserved and vital role in regulating SIS in response to cellular stress [11,70]. In *C. elegans*, heat stress-induced sleep requires ALA depolarization and the release of FMRFamide-like neuropeptides release encoded by the flp-13 gene [116]. Tea can regulate neurotransmitter levels [117,118]. Theanine is a derivative of glutamine that is structurally analogous to glutamate. It inhibits glutamine uptake in the glutamine-glutamate cycle via SLC38A1. This controls the balance of glutamate and glutamine in the brain to regulate sleep [119,120]. Additionally, theanine significantly increased the concentrations of acetylcholine and GABA and decreased the concentration of serotonin in the brain [87,121]. Polyphenols can prevent the reuptake of monoamine neurotransmitters and increase cerebral blood flow [122]. Aromatic substances, such as homeopathic jasmine, jasmine lactone, linalool oxide, and methyl jasmonate, significantly enhanced the expression of GABA receptors in Xenopus oocytes and increased the content of 5-HT and GABA in mouse brains [81,92].

#### 4.1.2. Tea Regulates Intestinal Flora to Mediate Sickness Sleep

The intestinal microbiota regulates host sleep and mental states through the microbiota-gut-brain axis [123]. In antibiotic-induced microbiota-depleted mice, the time spent in NREMS was reduced, while the number of REMS episodes increased; this was accompanied by frequent transitions from NREMS to REMS [124]. In addition, compared with healthy individuals, people with poor sleep quality have a higher relative abundance of *Firmicutes* and a lower relative abundance of *Bacteroidetes* in the intestinal flora [125,126,127]. Bioactive metabolites produced by the intestinal flora play an essential role in sickness sleep regulation. Lipopolysaccharide (LPS), lipoteichoic acid (LTA), and peptidoglycans induce sleep, fever, and anorexia. LTA may play a role in developing disease responses to gram-positive bacterial infections and in sleep signaling through the commensal gut microbiota [128].

Tea can effectively improve the proportion of beneficial to harmful microorganisms and regulate microbial diversity and metabolite production in the gut. Oolong, yellow, black, and dark teas have been reported to regulate intestinal microbiota. Black tea significantly promotes gastrointestinal transit and colonization of beneficial *Bifidobacterium*, *Lactobacillus*, and *Bacteroides* and inhibits the growth of harmful *Firmicutes*, *Escherichia coli*, and *Enterococcus* [129,130,131]. Yellow tea extracts altered the gut microbiota composition and increased community diversity and richness [132]. Green tea increases the abundance of *Flavonifractor plautii* (FP) in the gut microbiota, and LTA from FP was identified as the active component mediating IL-17 inhibition [133]. Tea compounds affect the growth of bacterial species involved in inflammatory processes, such as the release of LPS [134]. ECg promotes the release of LTA from the plasma membrane of *Staphylococcal* cells [135]. L-theanine improved intestinal dysbiosis by decreasing the ratio of *Firmicutes*/*Bacteroidetes*, along with increased fecal SCFA concentrations [136].

### 4.2. Tea Acts on Damage Repair to Mediate Sickness Sleep

Damage repair is considered a biological function of SIS. Sleep regulates repair mechanisms and immune responses in the body and promotes neural repair, metabolite clearance, and circuit reorganization [137]. In addition, sleep affects both humoral and cellular immunities. In mice, it significantly increases the number of monocytes in the blood and spleen and enhances the ability of monocytes and neutrophils to produce reactive oxygen species (ROS) [138]. Recent research on the interaction between sleep and damage repair has found that melatonergic regulators, which regulate circadian rhythms and sleep, inhibit the DNA damage response and activate the RAS/MAPK signaling pathways [139]. The antimicrobial peptide (AMP) neuropeptide-like protein (NLP)-29 in *C. elegans* acts through the neuropeptide receptor NPR-12 in locomotion-controlling neurons RIM and PVC, which are presynaptic to RIS neurons and depolarize this to induce sleep [140].

#### 4.2.1. DNA Damage Repair

External carcinogens and endogenous cellular processes cause DNA damage. The major endogenous sources of DNA damage are errors in DNA replication and spontaneous chemical changes produced by ROS, carbonyl stress, and hydrolysis of the glycosylic bonds. Exogenous damage is caused by exogenous agents, such as ultraviolet radiation, ionizing radiation, and various chemicals [141,142,143,144]. In long-term evolution, organisms have developed a series of DNA damage repair mechanisms, such as DNA repair and DNA damage response (DDR), to maintain genetic integrity [142,145,146].

The components of tea may play a role in DNA damage repair mechanisms. Regular intake of tea polyphenols rescued UVB-induced miR-29 depletion and prevented tumor growth by maintaining reduced DNA hypermethylation [147]. EGCg and its degradation products ensure normal neuronal differentiation and synaptic growth by reducing DNA methylation of the synaptophysin promoter [103,104]. In addition, EGCg acts as an antioxidant that protects embryos from oxidative damage by restoring the expression of ribosome/tumor-related proteins [148]. The antibacterial activity of green tea catechins results from various mechanisms, including DNA damage [149].

Previous studies have found a strong association between DNA damage repair and sleep. For instance, in zebrafish and mice, the activity of the DDR initiator poly polymerase 1 (Parp1) increases following sleep deprivation, while the activity of the DDR proteins, Rad52 and Ku80, increases during sleep, revealing that DNA damage triggers sleep [150]. CEP-1, which encodes proteins in the DDR pathway, acts downstream or parallel to ALA activation to promote SIS in *C. elegans* [151]. Sleep also promotes the repair of DNA damage. Participants who worked overnight after sleep deprivation had a lower baseline DNA repair gene expression and more DNA breaks [152]. Sleep increases chromosome dynamics, which is necessary to reduce the number of DNA double-strand breaks in zebrafish [153].

#### 4.2.2. Immune Response

To maintain homeostasis, animals have evolved mechanisms that include physical barriers and behavioral and immune responses to defend against and eliminate pathogen infection [154]. During an immune response, various cytokines drive immune cells to sites of infection for clearance and repair. The inflammatory response to treatment with pro-inflammatory cytokines is mediated by p38 mitogen-activated protein kinase (MAPK) [155,156,157,158]. Systemic inflammation significantly affects metabolism and induces characteristic sleep responses [159]. Pro-inflammatory cytokines such as tumor necrosis factor-alpha (TNFα) and interleukin-1 (IL-1) are assumed to mediate increased sleep under inflammatory conditions [160]. TNFα knockout mice had increased REMS, and IL-1 receptor accessory protein (AcP) is required for NREMS [161].

Tea regulates pro-inflammatory cytokines during inflammatory responses. Studies have shown that Pu-erh tea and white tea act on the p38/MAPK pathway to mediate inflammatory responses [162,163,164]. Tea polyphenols can effectively alleviate inflammation by downregulating the level of TNFα-converting enzymes and reducing the expression of pro-inflammatory cytokines, such as IL-1β. L-theanine has been suggested to inhibit heat stress-induced imbalance in oxidative stress and inflammatory responses by reducing inflammatory factors, such as TNF-α, IL-6, and IL-1β, via the p38/MAPK pathway [84,113,114]. In addition, theanine reduced synaptic scaling by downregulating TNFα-induced AMPA receptor phosphorylation, which upregulated Homer1a expression, thereby improving sleep [19].

### 4.3. Tea Acts on Energy Metabolism to Mediate Sickness Sleep

The energy metabolism pathways mainly include lipid metabolism, adenosine monophosphate-activated protein kinase (AMPK), insulin/IGF-1 signaling (IIS), and the mammalian/mechanistic target of rapamycin (mTOR) signaling pathways. Studies have found that individuals with obesity showed lower fat oxidation and higher carbohydrate oxidative catabolism during sleep and experienced shorter sleep duration than normal-weight individuals. This indicates that energy metabolism and sleep are mutually regulated [165].

#### 4.3.1. Lipid Metabolism

Lipid metabolism can regulate sleep in living organisms. Brown adipose tissue and uncoupling proteins are essential for maintaining energy homeostasis and body temperature. Studies have shown that the pharmacological activation of brown adipose tissue promotes sleep. Uncoupling protein 1 (UCP1), which promotes thermogenesis in brown adipocytes, is necessary for increasing NREMS [166]. In *C. elegans*, the transcription factor ETS-5 promotes roaming and inhibits quiescence by regulating a complex network of serotonergic and neuropeptide signaling pathways through fat regulation [167].

Lipid metabolism is also regulated during sleep. Long sleep durations were significantly associated with low high-density lipoprotein (HDL) cholesterol levels [168]. Morning-to-evening-regulated pathways of carbohydrate and lipid metabolism are sensitive to sleep loss [169]. Lipid levels in *Drosophila* are altered considerably during sleep [159]. Recently, weight loss during sleep has become an important issue. Studies have found that sleep extension can significantly reduce energy intake and result in a negative energy balance to reduce weight [170].

Tea has been shown to regulate lipid metabolism. Yellow and oolong tea inhibits obesity by increasing energy expenditure and fatty acid oxidation [171]. In contrast, green, white, Fuzhuan, and raw Pu-erh tea inhibit fatty acid synthesis [171]. Among them, green tea leaf powder reduces body weight and total cholesterol in mice on a high-fat diet (HFD) by decreasing the expression of fatty acid synthase and sterol regulatory element binding protein-1c (SREBP-1c) [172]. Pu-erh tea treatment significantly reduces free fatty acid (FFA) synthesis and increases the expression of genes involved in FFA uptake and β-oxidation in HFD-induced obese mice [173,174]. L-theanine promotes the metabolic activity of brown adipose tissue and subcutaneous white fat by enhancing thermogenic gene expression [136].

#### 4.3.2. AMPK

AMPK, an AMP-activated protein kinase that controls cellular metabolic decisions, is activated by increasing the AMP/ATP ratio in the body under cellular stress, exercise, and hormones. AMPK mediates the interaction between energy and sleep. The AMPK/SIRT1/PGC-1α pathway regulates the expression of skeletal muscle clock genes and the circadian locomotor output cycle kaput (Clock) [175]. AMPK knockdown in neuropeptide leucokinin (Lk) neurons inhibits sleep in *Drosophila* [176]. Salt-inducible kinases (SIKs) are essential members of the AMPK family. A SIK family kinase 3 (SIK3) deletion mutation in a well-conserved protein kinase A (PKA) phosphorylation site, S551, caused a lethargic phenotype in mice, characterized by a reduced wake time and increased NREMS time and delta density [177,178]. In addition, a gain-of-function sleepy mutation in SIK3 can also increase NREMS power and amount [179]. KIN-29 in *C. elegans* is a homolog of SIK, which acts upstream of fat regulation and sleep-controlling neurons to transduce low cellular energy charges into the mobilization of fat stores, thus promoting sleep [180].

Tea acts on AMPK, which mediates energy metabolism and affects physiological activities, including lipid and glucose metabolism. Yellow and raw Pu-erh tea significantly upregulated AMPK (p-AMPK) in HFD-induced obese mice [171]. Green, yellow, and black tea combined with citrus, can activate the AMP-activated protein kinase (AMPK)/acetyl-CoA carboxylase (ACC) signaling pathway and upregulate the expression of p-AMPK, p-ACC, and CPT-1 proteins, thereby inhibiting fat accumulation [181]. The combination of white tea and jiaogulan significantly suppresses hepatic glucose 6-phosphatase (G6Pase) expression by activating the AMPK pathway, thereby inhibiting gluconeogenesis [162]. Catechins, which increase AMPK activity and reduce ACC activity in metabolic tissues, affect lipid metabolism by reducing triglyceride levels and lipid droplet formation [182,183].

#### 4.3.3. IIS Signaling Pathway

The insulin/IGF signaling pathway (IIS), an evolutionarily conserved hormonal pathway, comprises insulin, insulin-like growth factor (IGF), or insulin-like peptide, insulin receptor IR/IGFR, serine-threonine kinase AKT, and downstream target forkhead box O (FOXO) transcription factors. It is vital in regulating energy metabolism, growth, and stress resistance [184,185].

Tea can regulate energy metabolism by activating downstream targets through the IIS pathway. Tea polyphenolics reportedly possess blood glucose-lowering properties by improving insulin sensitivity [186]. One study has shown that green tea extract attenuates downstream signaling of the insulin-like growth factor receptor [187]. In *C. elegans*, complex I inhibition by EGCg and ECg induced a transient drop in cellular ATP levels and a temporary ROS burst, resulting in SKN-1 and FOXO/DAF-16 activation [188]. Linalool, an aroma substance in tea, activates downstream sod-3 and hsp-12.6 gene expression through FOXO/DAF-16 and affects fat accumulation in *C. elegans* [189]. Tea also regulates the IIS pathway to play an anti-oxidative stress role. Oolong tea extract enhanced IGF/IR/p-AKT mechanism in the IIS pathway to aid cellular adaptation against hypoxic challenges [47].

The IIS pathway plays a vital role in sleep regulation. In the treatment of patients with circadian rhythm sleep-wake disorders, improvements in symptoms were found to be strongly associated with increased serum concentrations of insulin-like growth factor (IGF-1) [190]. In-depth studies have found that the activity of orexin neurons is modulated by IGF-1, which controls sleep duration and architecture [191]. In *Drosophila*, insulin-like peptide 2 (DILP2) is required for starvation-induced changes in sleep depth [192]. The adipokinetic hormone (AKH)-FOXO pathway has been shown to respond to energy changes and adjust *Drosophila*’s sleep by remodeling the dorsal projections of the small ventral lateral neurons [193]. In *C. elegans*, RIS neurons are activated by the conserved insulin receptors IR/DAF-2 and FOXO/DAF-16, thereby inducing sleep to conserve energy during extended food deprivation [194,195].

#### 4.3.4. mTOR Pathway

The mTOR is a serine/threonine kinase. The mTOR signaling pathway, the downstream target of AKT in the IIS pathway, plays a vital role in integrating nutrition, energy metabolism, growth, and proliferation [196,197]. Recently, the mTOR signaling pathway was found to be closely related to sleep. Both mTOR activity and orexin expression were increased in the hypothalamic sections and cultured hypothalamic neurons of Tsc1GFAPCKO mice, which showed sleep disorders. Both sleep abnormalities and increased orexin expression restore mTOR activation [198]. Food deprivation can inhibit SIS through the target of rapamycin and transforming growth factor-β (TGF-β) nutrient signaling pathways in *C. elegans* [199].

Tea affects energy metabolism through the mTOR signaling pathway. Fuzhuan tea aqueous extract alleviated insulin resistance by activating the insulin signaling Akt/GLUT4, FOXO1, and mTOR/S6K1 pathways in the skeletal muscles [200]. Studies have shown that tea may regulate neuro-mechanisms through the mTOR signaling pathway. A study found that polyphenols significantly improve sleep deprivation-induced contextual memory deficits, possibly through activating the cAMP-response element binding protein (CREB) and mTOR signaling pathways [201]. Theanine can upregulate SLC38A1 expression to activate the intracellular mTOR signaling pathway required to replicate and form neurons and neuron orientation [202]. In contrast, tea polysaccharides repressed the proliferation of colon cancer line HCT116 cells by targeting lysosomes to induce cytotoxic autophagy, which might be achieved through mTOR- transcription factor EB (TFEB) signaling [203].

## 5. Conclusions and Remarks

This review summarizes the role and mechanisms of tea in regulating sickness sleep. The potential efficacy of tea in promoting recovery under stress is further discussed, and the mechanisms by which tea regulates sickness sleep and maintains energy homeostasis are postulated.

Firstly, the regulation of sickness sleep is related to the brain–gut axis as well as damage repair. Among them, the brain–gut axis directly regulates sickness sleep, and damage repair acts as a sleep motive to indirectly induce sickness sleep. Tea may have the potential to regulate sickness sleep with its vital function in relieving stress and regulating healthy sleep. However, current research on tea in sickness sleep is deficient, and it is unclear whether and how tea regulates sickness sleep. Based on the mechanisms of sickness sleep regulation and the health benefits of tea, possible mechanisms of tea modulation of sickness sleep have been proposed, including acting on the nervous system, regulating intestinal flora, mediating DNA damage repair, and immune response. Secondly, according to the role of sickness sleep in promoting energy homeostasis, this review discusses the role of tea in linking sickness sleep and energy homeostasis, with lipid metabolism and AMPK, IIS, and mTOR pathways as potential targets.

The role and regulatory mechanisms of sickness sleep are more extensively studied in lower organisms. Owing to the more complex regulatory mechanisms, the role of sickness sleep is still not well elucidated in higher organisms. Excessive sleepiness under stress or illness has been found to be detrimental to patient recovery; hence, exploring the role of tea in regulating sickness sleep cannot be too one-sided. Further animal experiments and clinical validation are still needed in the future to focus on systematically and comprehensively describing the positive and negative effects of tea on sleep regulation under different conditions. Negative emotions under stress and in illness, such as anxiety and depression, can affect biological sleep while being detrimental to health recovery. Theanine in tea acts as a sedative and calming agent, and aromatic substances are widely used as sleep aids. Therefore, it would be a promising application to develop the beneficial components of tea into a calming and tranquilizing product and use it for patient recovery.

Whether, and how, sickness sleep and energy homeostasis under stress are influenced by tea warrants further investigation. Mitochondria may be an entry point to explore this question. Owing to the intermediary role of mitochondria in energy metabolism and response to stress, research on how tea affects mitochondrial function during sleep regulation, including mitochondrial dynamics, mitochondrial autophagy, and mitochondrial biosynthesis, might be a great direction for future research.

Overall, new insights on tea regulation of sickness sleep will enrich the health benefits of tea, while the potential of tea to harmonize sleep and energy balance under stress will provide insights into improving disease treatment recovery. A warm cup of tea and a nap during stress or discomfort may provide enough energy for a better recovery.

## Figures and Tables

**Figure 1 foods-11-03930-f001:**
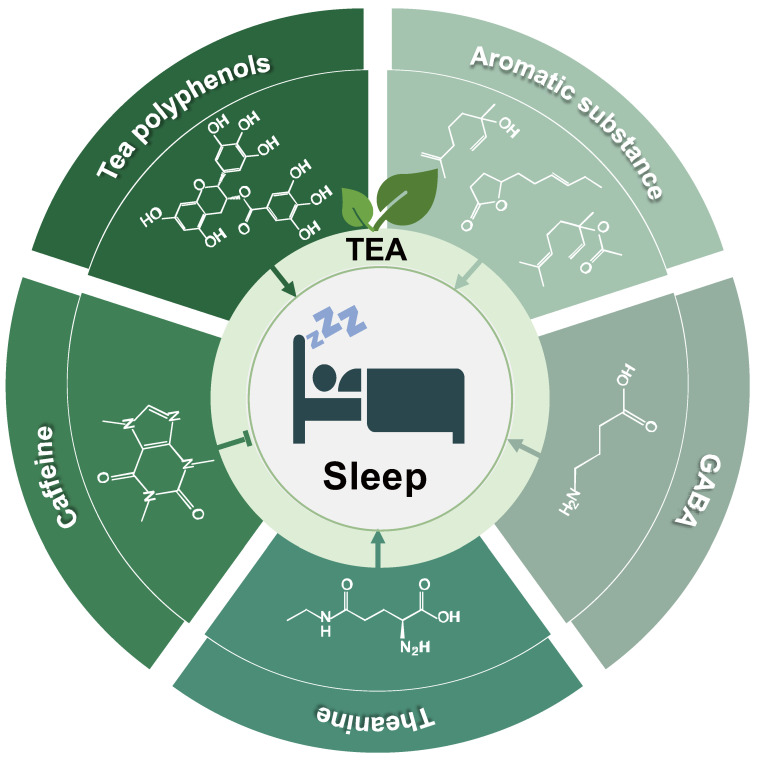
Main active ingredients of tea that affect sleep.

**Figure 2 foods-11-03930-f002:**
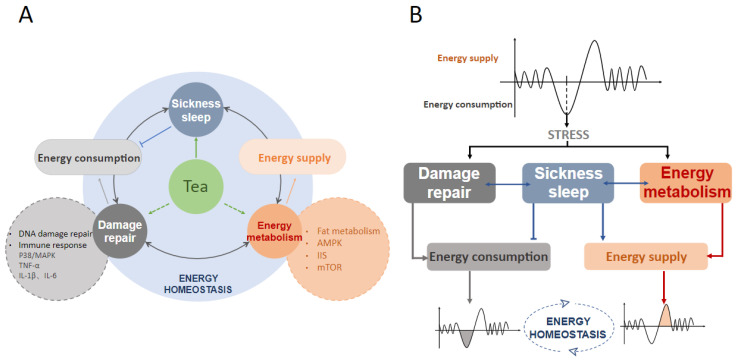
Tea coordinates sickness sleep, and energy homeostasis under stress. (**A**). Tea may act on damage repair and energy metabolism to promote sickness sleep and maintain energy homeostasis. (**B**). The energy homeostasis of animals is maintained under the dynamic balance of consumption and supply under stress. Sickness sleep, as a strategy for conserving energy consumption and optimizing energy allocation, helps to regulate damage repair and energy metabolism to maintain energy homeostasis under stress.

**Figure 3 foods-11-03930-f003:**
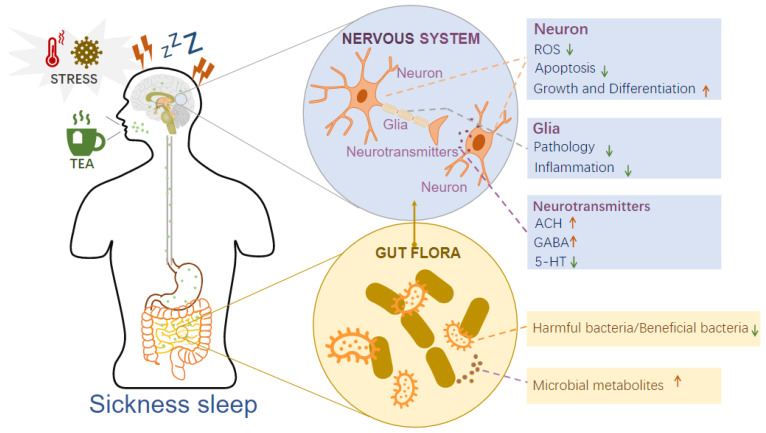
Tea regulates sickness sleep through the brain–gut axis. Components of tea act on the nervous system under stress. These components reduce neuronal ROS levels and cell apoptosis and promote neuronal growth and differentiation; inhibit glial pathology and reduce inflammation; and regulate the secretion of neurotransmitters, including ACH, GABA, and 5-HT. Furthermore, tea affects the intestinal flora, specifically by reducing harmful bacteria and promoting the production of beneficial bacteria metabolism.

**Table 1 foods-11-03930-t001:** The main health benefits of the six major tea types.

Types of Tea	Degree of Fermentation	Key Processing Technology	Major Health Benefits
Green tea	Non-fermented	Fixing	Antibacterial [34] Suppressing the amyloid beta levels and alleviating cognitive impairment in 5XFAD mice [35] Reducing lipid peroxidation and increasing total antioxidant capacity, and reducing oxidative damage [36] Significantly lowering the risk of developing liver cancer and improving the effect on body mass index, liver enzymes, and lipoprotein [37] Preventing obesity [38,39]
Black tea	Fully fermented	Fermentation	Exerting antibacterial activity against major periodontopathogens, attenuating the secretion of IL-8, and inducing hBD secretion in oral epithelial cells [40] Preventing radiation-induced increase of ACE activity and oxidative stress in the aorta [41] Limiting the formation of glycation products [42]
Yellow tea	Slightly fermented	Yellowing	Antioxidant and preventing gastric injury [43] Reducing blood glucose levels, increasing glucose tolerance, and preventing fatty liver in diabetes mice [44]
Oolong tea	Semi-fermented	Rotating	Neurodegenerative and neurite outgrowth-promoting [45] Inhibiting cancer cell proliferation [46] Providing cardio-protective benefits during hypoxic conditions [47] Prolonging lifespan and improving health span by curtailing the age-related decline in muscle activity and the accumulation of age pigment in *C. elegans* [48]
White tea	Slightly fermented	Withering	Inhibiting PhlP-induced aberrant crypt foci by altering the expression of carcinogen-metabolizing enzymes in rats [49]
Dark tea	Post-fermented	Pile fermentation	Decreasing risks of coronary heart disease and diabetes [50] Scavenging of DPPH and ABTS free radicals [51] Regulating the glycolipid metabolic disorders [52] Decreasing body weight and serum triglycerides for SD rats [53]

## Data Availability

The data that support the findings of this study are available from the corresponding author upon reasonable request.

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
