# Peer review of "New Perspectives on Sleep Regulation by Tea: Harmonizing Pathological Sleep and Energy Balance under Stress"

_foods, 2022, doi:10.3390/foods11233930_

Round 1

Reviewer 1 Report

The manuscript entitled “Tea coordinates sickness sleep and energy homeostasis under 2 stress: potential mechanisms speculated” is an interesting literature review on tea as a beverage rich in bioactive compounds with beneficial health properties. The manuscript is well written and structured with interesting figures, however, it should be improved.

- I consider that the title of the manuscript should be revised and improved in order to become more captivating and direct.

- The abstract is the most important part of the study in the sense that it is the presentation of the article and that can define whether the manuscript is read or not. In this way, I consider that the abstract presented is too reductive.

- The introduction is also reductive. In the end, the innovation and novelty of this bibliographic review should be highlighted, as well as its importance for the scientific community.

- In point 1.1. intend to address the review on tea. There would be so much more to explore at this point! In particular, distinguish the types of tea. As a review, it would be interesting to add at this point a table exploring the different types of tea and the respective benefits studied in the literature.

- The quality of the figures presented in the manuscript needs to be improved

- Point 3.1 needs to be greatly improved.

- Point 4 Summary and Outlook being a conclusion point has to be further explored highlighting the importance/impact of this review. Expose what has already been studied and what remains to be explored.

- Add information in “Author Contributions”, “Data Availability Statement” and “Conflicts of Interest”

Reviewer 2 Report

This study utilized the function of tea in modulating energy homeostasis among sickness sleep, energy metabolism, and damage repair in model organisms. This study suggested a potential mechanism of tea regulating sickness sleep. Overall, this study addresses a topic of high relevance for research and also for practice. However, I believe some issues need revision and clarification. 

Reviewer 3 Report

Dear Authors,Thank you very much to Editor for inviting me to review your publication. Tea is one of the most widely drunk drinks worldwide. Moreover, it has many positive effects on the human body, confirmed by the literature. Congratulations to the authors for evaluating the effects of tea consumption on the vital physiological functions of the body, which undoubtedly include sleep.

Below are my suggestions / comments:
Line 38: dot before
reference

line 244: dot before and after the reference

References are in bold and normal format - remove bold or make it uniform throughout the document.

There should be a space between the text and the reference (the entire document).

The conclusions section is missing. The last sentence could be found in it.

Round 2

Reviewer 1 Report

All requested changes have been taken into account and I suggest their acceptance

Author Response

Thank you for your guidance, we have accepted all revisions.

Reviewer 2 Report

Most of my remarks from the previous round of revision have been addressed. Yet, there are some issues need to be addressed. 
